# Association between lower limb alignment and low back pain: A systematic review with meta-analysis

**Saeedeh Abbasi**[ORCID]**, Seyed Hamed Mousavi**[ORCID]*****, Fateme Khorramroo**[ORCID]

Department of Sport Injuries and Biomechanics, Faculty of Sport Sciences and Health, University of Tehran, Tehran, Iran

* musavihamed@ut.ac.ir

**Data Availability Statement:** All data are presented in this study.

**Funding:** The author(s) received no specific funding for this work.

## Abstract

Low back pain (LBP) is a prevalent and costly condition globally, prompting the need to identify risk factors for effective management. Lower extremity misalignment plays a crucial role in the incidence of LBP. Therefore, we aimed to investigate the current evidence on a link between lower limb alignment and LBP, enhancing the understanding of this relationship. We searched four databases, including PubMed, Embase, Web of Science, and Scopus, up to September 2024. Inclusion criteria encompassed studies related to LBP and lower limb alignment, with eligible study types including case-control, cohort, and cross-sectional studies, all written in English. Two authors independently screened and assessed the methodological quality of the retrieved papers using the Downs and Black quality assessment checklist. Data of interest including study design, age, sample size, cases, association, and P-value were extracted from the included studies. Mean differences and 95% confidence intervals (CI) were calculated with random effects model in RevMan version 5.4. Thirteen articles evaluating lower limb alignment in individuals with LBP were included (102,359 participants in total). The meta-analysis results demonstrated that increased pronation with strong evidence(p = 0.02), increased hip internal rotation with moderate evidence, and increased knee internal rotation with limited evidence are associated with an increased risk of LBP. Overall, while some studies supported a relationship between lower limb alignment and LBP, the heterogeneity of study designs and methodological limitations hindered drawing a definitive conclusion. Future research should emphasize prospective cohort studies, incorporating objective measures of lower extremity alignment and standardized outcome measures.

## Introduction

Low back pain (LBP), a prevalent issue in the world, is a significant cause of functional limitation in individuals under 45 years of age [1, 2]. Studies have reported a prevalence of back discomfort ranging from 14.4% to 85% in various populations and groups [3]. LBP, generally characterized by pain and discomfort in the lumbar region, can be categorized based on

**Competing interests:** The authors have declared that no competing interests exist.

duration: acute, subacute, and chronic. LBP is often non-specific or mechanical in nature, with mechanical low back pain (MLBP) attributed to soft tissue, spinal, or intervertebral disc issues [4]. Chronic low back pain (CLBP) and acute low back pain (ALBP) are prevalent conditions that significantly impact individuals' quality of life and impose a substantial burden on healthcare systems worldwide. ALBP refers to pain persisting for less than six weeks, whereas CLBP is defined by pain lasting for 12 weeks or longer [5]. ALBP is often the result of mechanical issues or injuries such as muscle strains, ligament sprains, or disc herniations. These episodes are typically self-limiting, with many patients recovering within a few weeks with minimal intervention. However, the recurrence rate is high, and a significant proportion of acute cases can transition into chronic conditions if not managed appropriately [6, 7]. Since the kinetic chain concept highlights how dysfunctions in the lower limbs, such as poor posture, muscle imbalances, and altered lower limb alignment, can affect pelvic positioning, spinal posture, and upper body mechanics, this cascading effect can lead to musculoskeletal disorders [8].

Therefore, in order to comprehend the etiology of movement, impairments associated with CLBP and to develop effective strategies to prevent LBP, it is necessary to conduct studies that assess the biomechanical aspects of the lower limb complex during static and dynamic activities (walking). Hip transverse rotation range of motion (ROM) has been identified as a significant factor in LBP [9, 10] and limited hip ROM is associated with sagittal spinal misalignment, and this predisposes to LBP [11]. Compensatory excessive lumbar rotation due to limited pelvic rotation has been linked to microtrauma and subsequent LBP [6]. Changes in spinal curvature may be associated with alterations in the plantar arch, impacting the kinematic chains and their function during activities [12]. Therefore, it is crucial to evaluate foot arch alterations to prevent impairments. Several theories have been proposed regarding the etiology of central mechanical disorders, suggesting that they may be triggered or sustained by distal disturbances in the locomotor system, particularly in the lower extremities [13, 14]. Many clinicians functionally evaluate gait, pelvic leveling, firing patterns, muscle strength, and joint flexibility as a holistic composition of diagnostic factors that may inform patient care considerations related to LBP [15, 16]. In this context, three systematic reviews have examined the association between foot function and MLBP [17]. Additionally, the relationship between ankle and foot deviations such as flat foot, ankle instability, sagittal plane obstruction, and excessive pronation and their impact on LBP have been investigated in the literature [18]. Furthermore, the effects of foot posture and leg length discrepancies on LBP have also been the subject of study [19].

The available systematic review of the association between hip and knee kinematics and LBP is limited. Only a few studies have linked reduced ankle dorsiflexion to chronic and moderate to severe LBP in both sexes [18, 20]. Despite the scarcity of evidence, there is a connection between these joints' alignment and LBP. Most existing research has focused on the relationship between musculoskeletal factors, with a noticeable lack of knowledge on the effects of lower extremity malalignment. A systematic review is needed to explore the link between lower limb alignment and various types of back pain, aiming to understand the relationship between these factors. The aim of this article is to systematically review and analyze the existing literature on the relationship between lower limb alignment and LBP. By focusing on the biomechanical aspects of the lower limb complex during static and dynamic activities (walking), this review seeks to elucidate the role of hip, knee, and ankle kinematics in the etiology of LBP. Furthermore, this analysis aims to inform the development of targeted LBP prevention strategies for active populations.

## Materials and methods

### Research design

This systematic review will follow the Preferred Reporting Items for Systematic Reviews and Meta-Analyses (PRISMA) guidelines to ensure a comprehensive and standardized approach [21]. The study procedure was pre-registered on the International Prospective Register of Systematic Evaluation Procedures, PROSPERO CRD42023441515. The research design will consist of the following steps:

### Search strategy

A comprehensive search was conducted in PubMed, Scopus, Web of Science, and Embase, from database inception to September 2024. Key terms used in the search strategy were based on broad terms and related synonyms targeting two categories: ("lower limb" OR "lower extremity" OR "flatfoot" OR "flatfeet" OR "pes planus" OR "alignment" OR "hallux" OR "dorsiflexion" OR "foot posture" OR "valgus" OR "Varus" OR "range of motion" OR "rotation") AND ("back pain" OR "LBP" OR "lumbar pain" OR "spine"). To prevent missing any related studies, we also hand-searched all reference lists of eligible studies, related reviews, and meta-analyses.

### Eligibility criteria

All searches were conducted separately in accordance with pre-established inclusion criteria and extraction forms. Information was reviewed in the titles, abstracts, and full-text papers. The inclusion criteria were: "studies related to: LBP, lower limb alignment, adolescents, adults and active people. Types of studies include: randomized/nonrandomized, case-control, cohort, and cross-sectional. studies written in English". In this study, alignment refers to the positioning and orientation of body segments in relation to a defined reference line or anatomical landmarks that were measured under dynamic (walking) or static conditions. The exclusion criteria were: "non-English studies, studies not assessing the relationship between LBP and lower limb alignments, lower limb kinetic, LBP related to disk herniation or injury of the spine, sciatica, and limb amputation and treatment studies".

### Study selection

Two reviewers (S.A. and S.H.M.) independently screened the title, abstract and full-text of studies, in line with the inclusion criteria. If there were any disagreements, a consensus was achieved through discussion between two reviewers.

### Data extraction

Evidence profiles and summary of findings tables were created to summarize the findings and the quality of evidence for each outcome. All relevant data from the included studies were extracted by two authors (S.A & S.H.M) in June 2024. In instances where data was not available, we contacted the corresponding author to request the required information. We recorded the following data from each study: the first author's name and year of publication, study design, sex, mean age, comparison, tool, and result.

### Quality assessment

Two authors (S.A. and F.KH.) independently assessed the methodological quality of the included studies using the modified Downs and Black checklist. This checklist consists of 27

questions relating to quality of reporting (ten questions), external validity (three questions), internal validity (bias and confounding) (13 questions), and statistical power (one question) (S1 Table). Fourteen of them that are related to our study, marked yes (= 1) or no (= 0). When there was uncertainty or disagreement between the initial reviewers on how to score a particular question, the matter was referred to a third reviewer (S.H.M) for discussion (yes (= 2) or partially (= 1) and no (= 0)). Percentage agreement reliability between two reviewers is shown in Table 3. The overall quality of evidence was classified into four levels: high, moderate, and low [22].

## Synthesis of results

Mean differences and 95% confidence intervals (CI) were calculated with random effects model in RevMan version 5.4. A meta-analysis was conducted when a minimum of 2 studies examined the same outcome measure using similar methodologies. The level of statistical heterogeneity for pooled data was quantified by related P-values (P<0.05). Results were achieved by means of levels of evidence as defined by van Tulder et al. [23] and modified by Mousavi et al. [24] (Table 1).

# Results

Out of 32743 articles obtained in the initial search, 17,979 duplicates were removed. Additionally, 4,527 non-full-text and non-English articles were excluded. Following the screening of titles and abstracts, 10,117 unrelated studies and 65 articles with non-relevant designs (such as systematic reviews, non-randomized trials, or books) were eliminated. After full-text screening of 55 articles, 42 were excluded. The lists of excluded studies can be found in S3 Table. Ultimately, 13 studies met all inclusion criteria and were included in the systematic review [20, 25–36]. The PRISMA flow diagram illustrating the search process is presented in Fig 1.

## Studies characteristics

Table 2 Displays the characteristics of the studies that were included. A total of 13 studies with 102,359 participants, including 17,436 cases and 84,923 controls, were incorporated. These studies were published between 2006 and 2023. The age of the included research samples varied from 18 to 70. Of the 13 studies, five were cohort studies, four were cross-sectional, and the

**Table 1. Definitions of modified level of evidence.**

| Level of evidence | Description |
|---|---|
| Strong evidence | Pooled results from three or more studies, including a minimum of two high-quality studies which are statistically homogenous (p > 0.05)—may be associated with a statistically significant or non-significant pooled result. |
| Moderate evidence | Statistically significant pooled results from multiple studies, including at least one high-quality study, which are statistically heterogeneous (p < 0.05); or from multiple low- or moderate-quality studies which are statistically homogenous (p > 0.05); or statistically insignificant pooled results from multiple studies, including at least one high-quality study, which are statistically homogenous (p > 0.05). |
| Limited evidence | Results from multiple low- or moderate-quality studies that are statistically heterogeneous (p < 0.05); or from one high-quality study. |
| Very limited evidence | Results from one low- or moderate-quality study. |
| Conflicting evidence | Pooled results that are insignificant and from multiple studies, regardless of quality, that are statistically heterogeneous (p < 0.05, i.e., inconsistent). |

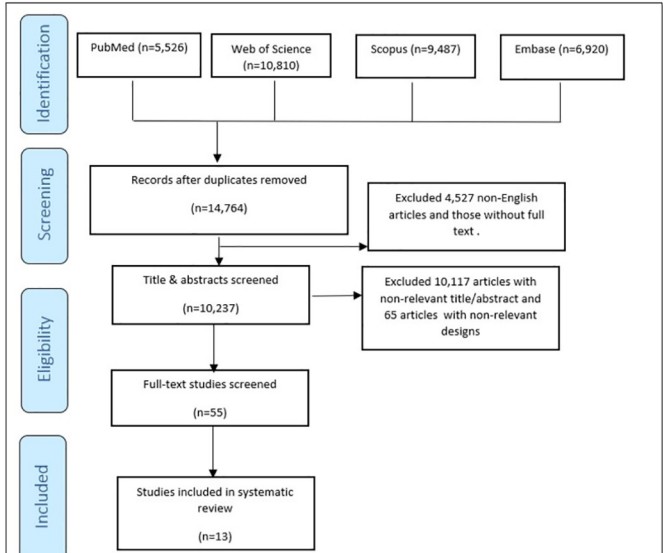

**Fig 1. Flow chart of study selection process.**

remaining studies were designed as case studies. Three studies enrolled only males, one enrolled only female, and others involved both sexes.

## Quality assessment

The results of quality assessments using the Downs and Black scales are presented (Table 3). We used fourteen questions of this scale to evaluate the quality of the included studies [14]; 11 studies were high quality and 2 studies were medium quality [32, 33]. Among the top scores, three studies scored 14 [25, 30, 34], two scored 13 [28, 35], and two others scored 12 [26, 27]. The other six studies scored 11 [29, 36], 10 [20, 31], 9 [33], 8 [32]. The eligible studies received an average score of 11.6.

## Ankle alignment and LBP

Nine studies investigated the differences in ankle alignment between groups with and without LBP [20, 25–28, 31–33, 36], and seven of these studies specifically explored the association between flat foot, navicular drop (ND), and calcaneal eversion with LBP [20, 25–28, 32, 36]. Additionally, three studies aimed to assess ankle ROM using motion analysis and goniometer measurements [28, 31, 33]. The focus of these studies was primarily on individuals with LBP. Two studies specifically examined ankle alignment during walking [31, 33]. In total, 16,952 individuals were included in these studies. Furthermore, six studies reported kinematic data for both men and women, two studies reported data for men only, and one study focused on women.

The findings of the meta-analysis investigating the correlation between ankle alignments and LBP are presented in Fig 2, Strong evidence indicates a significant association between flat foot and LBP (p = 0.02). Furthermore, an examination of flat foot severity (mild, moderate, and severe) in individuals with and without LBP revealed a notably higher prevalence of LBP in the moderate and severe pes planus groups compared to the mild and control groups [26]. However, a separate study that assessed flat feet using the ND did not identify any statistically

**Table 2. Data extraction of the included studies.**

| | Authors. year | Study | Participants | | | Tool /condition | Association | Outcomes (P-value) |
|---|---|---|---|---|---|---|---|---|
| | | | age | sample | case | | | |
| 1 | Menz et al. 2013 [20] | Cohort | 64.5±8.9 male | 863 male | 257 male | Arch index (2D) / static | Pes Cavus and LBP | 0.155 |
| | | | | | | | Pes Planus and LBP | 0.091 |
| | | | 63.4±8.9 female | 1067 female | 404 female | COP index (2D) /static | Supinated foot and LBP | 0.583 |
| | | | | | | | Pronated foot and LBP | 0.018 |
| 2 | Almutairi et al. 2021 [25] | Cross sectional | 27.3±8.8 | 844 male | 381 LBP | - /static | Flat foot and LBP | < 0.001 |
| | | | | 954 female | 255 LBP | | Flat foot and LBP | < 0.001 |
| 3 | Kosashvili et al. 2008 [26] | Retrospective | Adolescents | 78941 male | 15698 | Visual procedure Gould / static | Sever Pes planus and LBP | < 0.0001 |
| | | | | 18338 female | | | | |
| 4 | Brantingham et al. 2006 [28] | nonrandomized clinical study | 28.82 | 204 | 100 | 2D/ static | Decrease ankle dorsiflexion ROM and LBP | (Right) 0.002 |
| | | | | | | | | (Left) 0.032 |
| | | | | | | 2D/ static | Navicular drop and LBP | (Right) 0.005 |
| | | | | | | | | (Left) 0.324 |
| 5 | Harris-Hayes et al. 2009 [29] | Cohort | 26.96±7.74 | 35 male | 24 | Motion capture system (3D)/ Dynamic | LBP and hip transverse rotation ROM | 0.03 |
| | | | | 13 female | | | | |
| 6 | Rosenhagen et al. 2018 [30] | Case study | 14.7±2.3 | 836 | 166 | -/ static | Knee misalignment (valgum, varum) and LBP | < 0.05 |
| 7 | Rahimi et al. 2020 [31] | Cross sectional | 38.25±8.7 | 40 | 20 | Qualisys motion capture system, FDA (3D) / Dynamic | Less external rotation of dominant hip and LBP | 0 |
| | | | | | | | Less external rotation and Less abduction of nondominant hip and LBP | 0.04 |
| | | | | | | | Less flexion of dominant knee and LBP | 0.01 |
| | | | | | | | More plantar flexion of dominant ankle and LBP | 0.03 |
| 8 | Borges et al. 2013 [32] | Retrospective case series study | 30.45±6.25 | 18 female | 11 | SAPO Postural Analysis, Roman podoscope (2D) / static | High arch of foot and LBP | 0.0048 |
| | | | | | | | Flat foot and increasing the lumbar Curvature | |
| 9 | Brantingham et al. 2007 [27] | | | 58 | 30 | Ruler (2D) / static | Navicular drop and LBP | > 0.05 |
| | | | | | | Radiographic malleolar valgus index (2D) / static | Calcaneal eversion and LBP | |
| 10 | Farahpour et al. 2018 [33] | Case study | 26±2.9 | 45 male | 15 | Vicon MX motion analysis (3D) / Dynamic | Ankle inversion, knee flexion and internal rotation, hip internal rotation and LBP | < 0.05 |
| | | | | | | | Pronated foot and LBP | |
| 11 | Kim and Shin 2023 [34] | Cross sectional | 37.72±5.72 | 41 male | 25 | Formetric 4D device And Dual JTECH Inclinometer And goniometer/ static | Pelvic tilt imbalance, hip external rotation and LBP | 0 |
| | | | | | | | Pelvic tilt imbalance, hip internal rotation and LBP | 0.016 |
| 12 | Cejudo et al. 2020 [35] | Retrospective cohort | 22.5±2.89 | 13 male | 14 | -/ Dynamic | Hip transverse rotation and LBP | < 0.05 |
| | | | | 13 female | | | | |
| 13 | Bagwe and Varghese 2019 [36] | Cross sectional | 26.5±6.92 | 12 male | 36 | Goniometer and Ruler (2D) / static | Q angle and LBP | < 0.05 |
| | | | | 24 female | | | ND and LBP | |

All relevant data from the included studies were extracted by 2 authors (S.A & S.H.M) in June 2024. The authors confirm that all the aforementioned studies met the eligibility criteria for the systematic review.

**Table 3. Studies quality assessment based on Downs and Black checklist.**

| QUESTIONS | AIM CLEARLY DESCRIBED? | MAIN OUTCOMES DESCRIBED IN INTRODUCTION OR METHOD? | PATIENT'S CHARACTERISTICS CLEARLY DESCRIBED? | PRINCIPAL CONFOUNDERS CLEARLY DESCRIBED | MAIN FINDINGS CLEARLY DESCRIBED? | ESTIMATES OF RANDOM VARIABILITY PROVIDED FOR MAIN OUTCOMES? | P-VALUE REPORT FOR MAIN OUTCOME? | SUBJECTS ASKED TO PARTICIPATE REPRESENTATIVE OF SOURCE POPULATION? |
|---|---|---|---|---|---|---|---|---|
| Study/question number | 1 | 2 | 3 | 5 | 6 | 7 | 10 | 11 |
| Menz et al. 2013 [20] | 1 | 1 | 1 | 1 | 1 | 1 | 1 | 0 |
| Almutairi et al. 2021 [25] | 1 | 1 | 1 | 1 | 1 | 1 | 1 | 1 |
| Kosashvili et al. 2008 [26] | 1 | 1 | 1 | 1 | 1 | 1 | 1 | 0 |
| Brantingham et al. 2006 [28] | 1 | 1 | 1 | 2 | 1 | 1 | 1 | 0 |
| Harris-Hayes et al. 2009 [29] | 1 | 1 | 1 | 1 | 1 | 1 | 1 | 0 |
| Rosenhagen et al. 2018 [30] | 1 | 1 | 1 | 1 | 2 | 1 | 1 | 0 |
| Rahimi et al. 2020 [31] | 1 | 1 | 1 | 1 | 1 | 1 | 1 | 0 |
| Farahpour et al. 2018 [33] | 1 | 1 | 1 | 0 | 1 | 1 | 1 | 0 |
| Borges et al. 2013 [32] | 1 | 1 | 1 | 0 | 1 | 1 | 0 | 0 |
| Brantingham et al. 2007 [27] | 1 | 1 | 1 | 2 | 1 | 1 | 1 | 0 |
| Kim and Shin 2023 [34] | 1 | 1 | 1 | 1 | 1 | 1 | 1 | 1 |
| Cejudo et al. 2020 [35] | 1 | 1 | 1 | 1 | 1 | 1 | 1 | 1 |
| Bagwe and Varghese 2019 [36] | 1 | 1 | 1 | 1 | 1 | 1 | 1 | 0 |
| Percentage agreement reliability % | 100 | 100 | 92 | 96 | 100 | 96 | 100 | 96 |

| QUESTIONS | SUBJECTS PREPARED TO PARTICIPATE REPRESENTATIVE OF SOURCE POPULATION? | ANY DATA DREDGING CLEARLY DESCRIBED? * | APPROPRIATE STATISTICAL TEST PERFORMED? | OUTCOME MEASURES WERE RELIABLE AND VALID? | ALL PARTICIPANTS RECRUITED FROM THE SOURCE POPULATION? * | ALL PARTICIPANTS RECRUITED OVER THE SAME PERIOD OF TIME? | TOTAL |
|---|---|---|---|---|---|---|---|
| Study/question number | 12 | 16 | 18 | 20 | 21 | 22 | |

(*Continued*)

**Table 3.** (Continued)

| | | | | | | | | |
|---|---|---|---|---|---|---|---|---|
| 1 | Menz et al. 2013 [20] | 0 | 1 | 1 | 1 | 0 | 0 | 10 |
| 2 | Almutairi et al. 2021 [25] | 1 | 1 | 1 | 1 | 1 | 1 | 14 |
| 3 | Kosashvili et al. 2008 [26] | 0 | 1 | 1 | 1 | 1 | 1 | 12 |
| 4 | Brantingham et al. 2006 [28] | 1 | 1 | 1 | 1 | 0 | 1 | 13 |
| 5 | Harris-Hayes et al. 2009 [29] | 0 | 1 | 1 | 1 | 1 | 0 | 11 |
| 6 | Rosenhagen et al. 2018 [30] | 1 | 1 | 1 | 1 | 1 | 1 | 14 |
| 7 | Rahimi et al. 2020 [31] | 0 | 1 | 1 | 1 | 0 | 0 | 10 |
| 8 | Farahpour et al. 2018 [33] | 0 | 1 | 1 | 1 | 0 | 0 | 9 |
| 9 | Borges et al. 2013 [32] | 0 | 1 | 1 | 1 | 0 | 0 | 8 |
| 10 | Brantingham et al. 2007 [27] | 1 | 1 | 1 | 1 | 0 | 0 | 12 |
| 11 | Kim and Shin 2023 [34] | 1 | 1 | 1 | 1 | 1 | 1 | 14 |
| 12 | Cejudo et al. 2020 [35] | 0 | 1 | 1 | 1 | 1 | 1 | 13 |
| 13 | Bagwe and Varghese 2019 [36] | 0 | 1 | 1 | 1 | 1 | 0 | 11 |
| **Percentage agreement reliability %** | | 92 | 95 | 100 | 92 | 96 | 100 | |

1 = Yes; 0 = No.

*2 = Yes; 1 = Partially; 0 = No; * = the question discussed with the third reviewer.

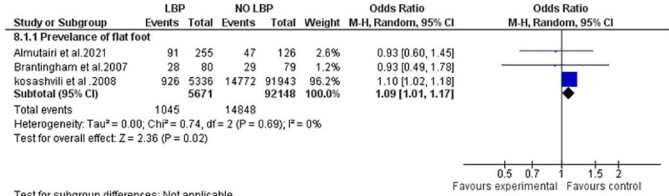

**Fig 2. Forest plot of the prevalence of flat foot in LBP individuals.**

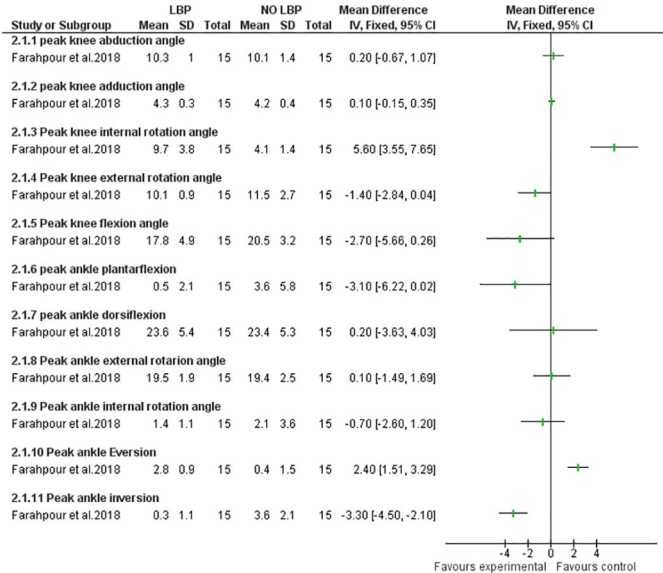

**Fig 3. Forest plot of the results of knee and ankle alignments in LBP individuals during the stance phase of walking.**

significant differences between subjects with and without LBP [27]. Fig 3, Illustrates the relationship between ankle ROM and LBP. Limited evidence suggests ankle eversion ROM was significantly higher, and ankle inversion was lower in the LBP group compared to the control. However, no significant differences were observed between the two groups in terms of other alignments (dorsiflexion, external rotation, etc.) (p < 0.05) [33].

## Knee alignment and LBP

Four studies investigated knee alignment in 237 individuals with and without LBP [30, 31, 33, 36]. These studies investigated knee function, with two using motion analysis for flexion/extension ROM [31, 33], one using radiography for valgus/varus [30], and another measuring the Q-angle [36]. Mainly centered on LBP, one study specifically explored knee alignment in non-specific LBP [36]. Two studies assessed knee alignment during walking, and sex-specific kinematic data were reported in some studies [31, 33]. Based on limited evidence, it found that peak knee internal rotation during walking was significantly greater in the LBP group compared to the control, while other LBP group alignments such as peak of knee external rotation, abduction, adduction and flexion had no difference with control (Fig 3). Additionally, the study revealed that knee internal rotation on the dominant side was less than on the non-dominant side, and individuals with LBP showed less knee flexion during late stance [31].

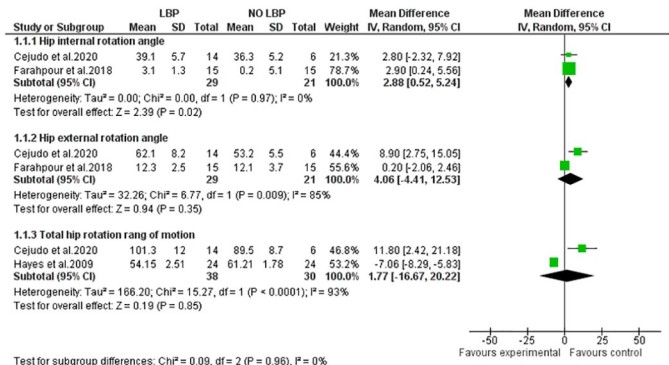

**Fig 4. Forest plot of the results of hip alignments in LBP individuals.**

## Hip alignment and LBP

Five studies investigated the differences in hip alignment between individuals with and without LBP [29, 31, 33–35]; The researchers assessed hip external and internal rotation ROM using goniometers, inclinometers, or motion analysis. Two studies focused on walking [31, 33], while another highlighted pelvic tilt imbalance as a contributing factor to hip transverse rotation ROM in the LBP group [34]. The total number of individuals examined across these studies was 98, with one study reporting data exclusively for males and the others including both sexes in their kinematic data. In Fig 4, Moderate evidence suggests that LBP is significantly associated with increased hip internal rotation ROM (P< 0.05), while no noticeable change was observed in hip external or total rotation between the two groups [29, 33, 35].

## Discussion

The aim of this research was to explore the connection between lower limb alignment and LBP, particularly concerning potential implications for preventive interventions, especially among adolescents and young adults. Misalignment of the lower extremities has been linked to heightened biomechanical stress, potentially contributing to pelvic instability and LBP [37, 38]. Various factors, such as repetitive movements, heavy lifting, unfavorable postures, and sports-related whole-body accelerations, may increase the risk of LBP through their impact on lower limb alignment. Understanding these associations holds promise for prevention and management strategies. This discussion aims to synthesize and analyze the findings of multiple studies to clarify the relationship between lower limb alignment and LBP in both adolescent and adult populations. A comprehensive analysis, involving 13 studies with 17,436 participants, reveals significant associations. Meta-analysis results indicate a significant connection between the prevalence of flat feet and LBP (p = 0.02). Furthermore, individuals with LBP exhibit a significantly increased hip internal rotation ROM (P < 0.05), but there is no significant difference in total hip transverse rotation between both groups. Additionally, more ankle eversion and peak knee internal rotation were observed as contributing factors.

### Foot and ankle alignment and LBP

The interaction between the body's back and feet through the lower extremity kinematic chain is highlighted in seven studies exploring ankle alignment, including flat foot, ankle dorsiflexion ROM, and pronation during static or dynamic activities [20, 25–28, 35, 36]. Studies consistently link a decreased medial longitudinal foot arch with a higher prevalence of LBP. Meta-

analysis (Fig 2) provides strong evidence, revealing a greater prevalence of flat foot in the LBP group compared to healthy individuals. This finding is supported by a retrospective study with 97,279 subjects [26]. Additionally, during the stance phase of walking, LBP individuals exhibited less inversion and more eversion in peak ankle alignments compared to healthy individuals.

A substantial retrospective study examined the associations of foot posture (pes cavus/pes planus) and foot function (supinated/pronated) with LBP in both sexes [20]. It reveals that individuals with flat feet have a higher prevalence of LBP, especially among overweight, older, and occupied females [25]. Pronated foot function was significantly associated with a higher risk of LBP in females compared to normal foot function. While no association was found between static foot position and LBP [20], excessive foot pronation was more likely to affect proximal structures in women due to sex differences in plantar alignment, ROM, and lower limb and vertebral joint function [39–42]. As such, kinematic changes resulting from excessive foot pronation may be more easily transmitted to proximal structures in women.

Moreover, the prevalence of LBP was significantly higher in both moderate and severe pes planus individuals. Notably, the prevalence of rigid pes planus was three times higher in males than females (0.9% vs. 0.3%), possibly due to the fact that although females have more joint mobility and ligament laxity in the foot compared to males, which leads to a more dynamic internal longitudinal arch. But females rely more on the toe flexor muscles to support the arch, which may explain the difference in arch stability between the sexes [43–45]. In both sexes, moderate and severe pes planus correlate significantly with anterior knee pain and LBP at varying rates [26]. A study clearly demonstrated a significant correlation between alterations in the medial longitudinal arch and changes in lumbar curvature that cause LBP. Smaller angulations in the lumbar spine (lumbar hyper lordosis) were observed in individuals with larger midfoot footprints [46]. Individuals with LBP tend to exhibit decreased ankle dorsiflexion ROM in both feet, and stiffness in the ankle and great toe [28]. Flat feet can contribute to locomotor issues, leading to altered gait, misalignment, and reduced shock absorption, potentially worsening LBP [47]. At heels strike, when the foot begins to pronate, the calcaneus inverts while the talus adducts and the ankle plantar flexes [48]. Medial displacement of the talus causes corresponding medial rotation of the tibia, which in turn leads to medial rotation of the femur [46]. In theory, an increase in internal rotation of the femur results in an anterior tilt of the pelvis due to the tight fibrous connection provided by the sacroiliac joint [49]. Proper ankle alignment ensures even weight distribution and ground reaction force (GRF) in the lower extremity. A study on LBP revealed ankle motion patterns, indicating a more plantarflexed position early in the stance phase and reduced dorsiflexion later in the stance phase in the LBP group [31]. This aligns with a previous report of increased stiffness of plantar flexors in people with LBP [50].

Knee and ankle sagittal movement patterns may help reduce lower back mechanical load by avoiding downward displacement. Further research is needed to understand this relationship. Clinicians should assess ankle alignment, incorporate gait analysis, and use interventions like orthotic devices, footwear, exercises, and gait training.

## Knee alignment and LBP

The findings underscore the significance of evaluating knee misalignment as a potential contributing factor to the development of LBP. This highlights the imperative for proactive measures and targeted interventions aimed at addressing knee misalignment in young individuals, with the objective of averting the onset of LBP within this demographic [30]. Deviations in knee alignment may lead to muscle imbalances in the lower extremities and the surrounding

musculature of the spine, potentially contributing to LBP [51]. While our meta-analysis didn't show statistically significant differences in knee alignment, we did find limited evidence indicating a notable increase in internal rotation peak during walking stance in individuals with LBP compared to those without.

A cross-sectional study confirmed the association between limited ROM in the knee joint and tightness in the quadriceps muscle with LBP [52]. Similarly, Rahimi et al. observed a distinct pattern of knee sagittal motion during the stance phase in the LBP group, with a less flexed knee during late stance, potentially linked to hamstring weakness. Furthermore, it was found that knee internal rotation on the dominant side during the stance phase is less than on the non-dominant side [31]. Previous studies have indicated that GRF on the dominant side is lower than on the non-dominant side in individuals with LBP, which could potentially explain the observed asymmetries [28]. Another study revealed that individuals with both foot pronation and LBP exhibited increased knee internal rotation during the stance phase [33]. Further research should continue to explore these associations, including prospective studies aimed at establishing causality and the underlying mechanisms. Additionally, investigating the effects of interventions such as orthotics or corrective exercises on knee alignment can provide valuable guidance to clinicians in the prevention and management of LBP.

## Hip alignment and LBP

Recent research links hip alignment with LBP, indicating a biomechanical connection due to the proximity of the hip and lumbopelvic region. Meta-analysis (Fig 4) strongly supports reduced hip internal rotation in individuals with LBP, while external and total transverse rotation show no significant differences. Limited hip internal rotation range of motion, often caused by muscle tightness, joint stiffness, or structural abnormalities, may lead to compensatory movements in the lumbopelvic region. These compensations during activities requiring hip transverse rotation can increase tissue loading, causing microtrauma, LBP symptoms, and potentially even macro trauma [33, 35].

A cross-sectional study reported lower hip transverse rotation ROM and differing right and left foot hip transverse rotation ROM in individuals with LBP compared to those without LBP [29]. Similarly, Rahimi et al. observed a notably different hip transverse rotation pattern on the dominant side during the stance phase in individuals with LBP, with reduced external rotation early in stance and decreased internal rotation later in stance. On the non-dominant side, the LBP group exhibited distinct patterns of hip transverse and frontal motion during the stance phase, with reduced external rotation and abduction. They also reported an association between increased hip external rotation and hip total rotation with LBP [35, 36]. Another recent study found that changes in pelvic tilt are linked to alterations in hip internal/external rotation, which are identified as risk factors for LBP [34].

These findings underscore the importance of considering lower extremity alignment, functions, and asymmetries in the assessment and management of LBP, particularly in athletes. Addressing misalignments and improving the ROM may aid in reducing the risk of LBP and enhancing overall biomechanics. However, further research is necessary to establish a more comprehensive understanding of this relationship.

## Limitations and research implications

The reliance on various static measurement indexes in the included studies significantly restricts the capacity to reach definitive conclusions and apply findings across different clinical contexts. A consensus must be reached on the use of a standardized static measurement index to improve comparability and generalizability. Moreover, neglecting to account for sex and

pain duration in the data analysis of some included studies overlooks significant differences in treatment responses, potentially invalidating conclusions for various populations. Researchers should consistently incorporate sex differences and pain duration categories to improve insights and relevance. Inconsistent measurement methods and terminology (e.g., pes planus, pronation, ND, ankle ROM) create ambiguity that complicates result comparisons across studies, undermining the reliability of evidence. Standardizing biomechanical factors and terminology is crucial for enhancing clarity and facilitating reliable comparisons.

## Clinical implications

The clinical implications of this research suggest the importance of assessing patients with LBP to identify specific types and associated disturbed lower extremity locomotor factors. This assessment aims to determine the suitability and potential benefits of intervention for each individual. Orthotics may play a beneficial role in increasing shock absorption and decreasing LBP, independent of raising the arch. Additionally, controlling foot motion through the application of wedges to the sole of the foot may lead to an earlier onset of erector spine muscle activity when walking, potentially correcting the delayed activity of this muscle observed in LBP individuals and aiding in controlling the motion of the trunk. Considering the complex and multifactorial nature of the association between lower extremity (such as ankle and hip) alignments and LBP, clinicians should still consider lower extremity alignment as one of the important factors in the management of LBP. Future research should explore the role of lower extremity alignment in managing low back pain, optimize interventions, and investigate longitudinal studies for sustained improvements. Additionally, it should consider assessing dynamic alignment alongside static alignment for personalized treatment strategies.

## Conclusion

In conclusion, the research indicates that lower limb alignment, particularly in the hip, knee, and ankle, significantly impacts LBP. The included studies focus on ankle alignment, with findings showing that flat feet and a decreased longitudinal arch of the foot are associated with an increased prevalence of LBP. Stiffness in the ankle or big toe and pelvic unleveling from flat feet also contribute to altered gait and alignment, leading to LBP. Interventions targeting lower extremity locomotor factors, such as orthotics, may benefit LBP individuals. Additionally, knee misalignment, including valgus/varum, limited knee flexion, and asymmetries between dominant and non-dominant sides, increases the risk of developing LBP. Increased hip internal or external rotation can also lead to compensatory lumbopelvic rotation, which is a risk factor for LBP. Overall, addressing lower limb alignment, especially in relation to flat feet and hip misalignment, is crucial to managing and preventing LBP. Further research is needed to explore interventions aimed at improving lower limb alignment and reducing the prevalence of LBP.

## Supporting information

**S1 Table. Down and Black scale (quality assessment).**
(DOCX)

**S2 Table. Prisma checklist.**
(DOCX)

**S3 Table. Table of the excluded articles.**
(CSV)

## Author Contributions

**Conceptualization:** Saeedeh Abbasi, Seyed Hamed Mousavi, Fateme Khorramroo.

**Formal analysis:** Seyed Hamed Mousavi, Fateme Khorramroo.

**Methodology:** Saeedeh Abbasi, Seyed Hamed Mousavi.

**Project administration:** Saeedeh Abbasi, Fateme Khorramroo.

**Resources:** Saeedeh Abbasi.

**Supervision:** Seyed Hamed Mousavi.

**Validation:** Fateme Khorramroo.

**Writing – original draft:** Saeedeh Abbasi.

**Writing – review & editing:** Saeedeh Abbasi, Seyed Hamed Mousavi.

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
