## [Decision Letter · Decision Letter 0]

12 Aug 2024

PONE-D-24-23407Is lower limb alignment associated with low back pain?: A systematic review with meta-analysisPLOS ONE

Dear Dr. Abbasi,

Thank you for submitting your manuscript to PLOS ONE. After careful consideration, we feel that it has merit but does not fully meet PLOS ONE’s publication criteria as it currently stands. Therefore, we invite you to submit a revised version of the manuscript that addresses the points raised during the review process.

We look forward to receiving your revised manuscript.

Kind regards,

Mehrnaz Kajbafvala, Ph.D

Academic Editor

PLOS ONE

Journal Requirements:

2. In the online submission form, you indicated that [The data underlying the results presented in the study are available from (Saeideh.abbasi@ut.ac.ir).]. 

Reviewers' comments:

Reviewer's Responses to Questions

**Comments to the Author**

1. Is the manuscript technically sound, and do the data support the conclusions?

Reviewer #1: Yes

Reviewer #2: Yes

2. Has the statistical analysis been performed appropriately and rigorously? 

Reviewer #1: Yes

Reviewer #2: I Don't Know

3. Have the authors made all data underlying the findings in their manuscript fully available?

Reviewer #1: Yes

Reviewer #2: Yes

4. Is the manuscript presented in an intelligible fashion and written in standard English?

Reviewer #1: Yes

Reviewer #2: Yes

5. Review Comments to the Author

Reviewer #1: This study was aimed to investigate the current evidence on a link between lower limb alignment and LBP. The meta-analysis results demonstrated that increased pronation with strong evidence(p=0.02), increased hip internal rotation with moderate evidence, and increased knee internal rotation with limited evidence are associated with an increased risk of LBP. Overall, while some studies supported a relationship between lower limb alignment and LBP, the heterogeneity of study designs and methodological limitations hindered drawing a definitive conclusion. Overall, the study is interesting. However, a little correction is needed.

Comment#1

Keywords. Please add LBP to Keywords.

Comment#2

Methods. Please update search strategies.

Reviewer #2: This review was on “lower limb alignment in patients with low back pain”. The manuscript was generally well written but I came to the opinion that this article needs major revision. I am including my comments for your reference. I hope authors find this information helpful.

Title

Please revise title so that it more accurately reflects the review. Avoid titles written as questions.

Reformat – do not make as a question

Abstract

• The manuscript needs a structured abstract with distinct headings.

Introduction

• Page 1, lines 34-35: " ALBP refers to pain persisting for less than six weeks, whereas CLBP is defined by pain lasting for 12 weeks or longer." The sentences need valid citation. Please add.

• Page 1, lines 39-40: " LBP is caused by poor posture, muscle imbalances, and altered lower limb alignment, influenced by the complex interplay of pelvis, hip, knee, and ankle joints.” According to previous studies there is no causal relationship between these factors and LBP. Please revise.

• Because the authors excluded “treatment studies” second objectives of the study must be stated clearly. Please revise.

Methods

• How the alignment as a variable was defined? In what conditions? Dynamic or static? define all the methods used to decide which objective results for alignment to collect.

• I would like to see more information about the methods used to assess risk of bias for different studies included in this review. What method used for appraising different studies?

• Page 3, lines 96-97: “Two authors…independently assessed the methodological quality of the included studies using the modified Downs and Black checklist with high reliability (0.89) and validity (0.90).” % agreements and kappa values are usually reported for raters in review studies. It is not clear why validity index was reported in this review. Please comment.

Results

• Page 8, lines 164-165: “Two studies assessed knee alignment during walking, and sex specific kinematic data were reported in some studies.” The sentence needs valid citation. Please add.

Discussion

• The results for alignment in dynamic and static conditions were presented. It would be good to “synthesize and analyze” the role of these two conditions and LBP separately. Can they further comment on this?

• Page 11, lines 214-215: “…possibly due to increased soft tissue elasticity in females.” The sentence needs valid citation. Please add.

• The section for limitation must be revised. Discuss the factors that limit the generalizability of study results.

• Page 12, lines 285-287: “Considering the complex and multifactorial nature of the relationship between lower extremity alignment and LBP, clinicians should take alignments and targeted interventions into account in the management of LBP.” Based on the results of this review there were no causal relationship between alignment and LBP. Please revise.

• The implications of the findings of the study for future research should also be clearly stated.

6. PLOS authors have the option to publish the peer review history of their article (what does this mean?). If published, this will include your full peer review and any attached files.

Reviewer #1: No

Reviewer #2: No

---

## [Author Response · Author response to Decision Letter 0]

21 Aug 2024

Dear editor and reviewers;

We thank you for your insightful and constructive suggestions. Based on your helpful suggestions, we were able to further improve our manuscript. We carefully considered and addressed all your specific comments and revised the text if necessary. 

Please find your comments in bold and our responses in italic font. In the manuscript, the highlighted parts in green are the revisions made based on the comments of reviewer1 and the highlighted parts in yellow are the revisions made based on the comments of reviewer2. 

In the online submission form, you indicated that [The data underlying the results presented in the study are available from (Saeideh.abbasi@ut.ac.ir).]. 

Thank you for your comment. Edited as bellow:

All data are presented in this study.

Reviewer #1

Abstract 

1* Keywords. Please add LBP to Keywords.

Thank you for your comment. Edited as bellow:

Keywords: low back pain, postural alignment, lower extremity, biomechanics, knee, ankle, hip

 Methods

2* Please update search strategies.

Thank you for your notice. 

It had already been updated to May 2024 in the Abstract but not in the Methods section. Now it is corrected as bellow in the Methods section too:

A comprehensive search was conducted in PubMed, Scopus, Web of Science, and Embase, from database inception to May 2024. 

Reviewer #2

Thank you for taking the time to provide detailed feedback on our manuscript. We appreciate your insights and constructive criticism. Your comments will be invaluable in refining our study. We carefully address each of the points you have raised to enhance the clarity of our rationale, align the benefits with the actual outcomes, improve the accuracy of our results, and enhance the structure of our discussion. Your input will undoubtedly contribute to the overall quality of our work. 

 Title 

1* Please revise title so that it more accurately reflects the review. Avoid titles written as questions. Reformat – do not make as a question

Thank you for your comment. Edited as bellow:

Association between lower limb alignment and low back pain: A systematic review with meta-analysis

2* The manuscript needs a structured abstract with distinct headings.

This section is written based on the PLOS ONE style templates: https://journals.plos.org/plosone/s/file?id=wjVg/PLOSOne_formatting_sample_main_body.pdf. 

But if it is necessary we will change the format.

Introduction

3* Page 1, lines 34-35: " ALBP refers to pain persisting for less than six weeks, whereas CLBP is defined by pain lasting for 12 weeks or longer." The sentences need valid citation. Please add.

Thank you for your comment. We added a research paper to the reference:

ALBP refers to pain persisting for less than six weeks, whereas CLBP is defined by pain lasting for 12 weeks or longer (5).

4* Page 1, lines 39-40: " LBP is caused by poor posture, muscle imbalances, and altered lower limb alignment, influenced by the complex interplay of pelvis, hip, knee, and ankle joints.” According to previous studies there is no causal relationship between these factors and LBP. Please revise.

Thank you for your comment. We clarified as bellow:

Since the kinetic chain concept highlights how dysfunctions in the lower limbs, such as poor posture, muscle imbalances, and altered lower limb alignment, can affect pelvic positioning, spinal posture, and upper body mechanics, this cascading effect can lead to musculoskeletal disorders.[8].

5* Because the authors excluded “treatment studies” second objectives of the study must be stated clearly. Please revise.

Thank you for your comment. The following sentence was removed since it was not in the scope of our research.

“Thereby enhancing the understanding of movement impairments helps to develop effective treatment strategies.”

 Methods

6* How the alignment as a variable was defined? In what conditions? Dynamic or static? define all the methods used to decide which objective results for alignment to collect.

Thank you for your comment. We clarified as bellow:

In this study, alignment refers to the positioning and orientation of body segments in relation to a defined reference line or anatomical landmarks that were measured under dynamic or static conditions.

Also, we have added a ‘Tool/ condition’ column to the relevant Table2, specifying which alignments were measured and under which conditions, whether dynamic or static. These revisions aim to provide a clearer understanding of the alignment variable used in the studies reviewed.

7* I would like to see more information about the methods used to assess risk of bias for different studies included in this review. What method used for appraising different studies? 

Thank you for your comment. We clarified and edited as bellow:

Two authors (S.A. and S.H.M.) independently assessed the methodological quality of the included studies using the modified Downs and Black checklist. This checklist consists of 27 questions relating to quality of reporting (ten questions), external validity (three questions), internal validity (bias and confounding) (13 questions), and statistical power (one question) (S1. Table). Fourteen of them that are related to our study, marked yes (=1) or no (=0). When there was uncertainty or disagreement between the initial reviewers on how to score a particular question, the matter was referred to a third reviewer (F.KH) for discussion (yes (=2) or partially (=1) and no (=0). Percentage agreement reliability between two reviewers is shown in Table 3. The overall quality of evidence was classified into four levels: high, moderate, and low [22].

8* Page 3, lines 96-97: “Two authors…independently assessed the methodological quality of the included studies using the modified Downs and Black checklist with high reliability (0.89) and validity (0.90).” % agreements and kappa values are usually reported for raters in review studies. It is not clear why validity index was reported in this review. Please comment.

Thank you for your comment. Since reliability and validity of this scales aren’t usually reported, we removed it and clarified as bellow:

Two authors (S.A. and S.H.M.) independently assessed the methodological quality of the included studies using the modified Downs and Black checklist. This checklist consists of 27 questions relating to quality of reporting (ten questions), external validity (three questions), internal validity (bias and confounding) (13 questions), and statistical power (one question) (S1. Table). Fourteen of them that are related to our study, marked yes (=1) or no (=0). When there was uncertainty or disagreement between the initial reviewers on how to score a particular question, the matter was referred to a third reviewer (F.KH) for discussion (yes (=2) or partially (=1) and no (=0). Percentage agreement reliability between two reviewers is shown in Table 3. The overall quality of evidence was classified into four levels: high, moderate, and low [22].

Result

9* Page 8, lines 164-165: “Two studies assessed knee alignment during walking, and sex specific kinematic data were reported in some studies.” The sentence needs valid citation. Please add.

Thank you for your comment. We added two references related to the sentence.

Two studies assessed knee alignment during walking, and sex-specific kinematic data were reported in some studies (26,36).

Discussion

10* The results for alignment in dynamic and static conditions were presented. It would be good to “synthesize and analyze” the role of these two conditions and LBP separately. Can they further comment on this?

Thank you for your comment. Static and dynamic conditions were separated in the results for meta-analysis. As requested, we discussed them separately in the discussion.

11* Page 11, lines 214-215: “…possibly due to increased soft tissue elasticity in females.” The sentence needs valid citation. Please add.

Thank you for your comment. We edited as bellow:

possibly due to the fact that although females have more joint mobility and ligament laxity in the foot compared to males, which leads to a more dynamic internal longitudinal arch. But females rely more on the toe flexor muscles to support the arch, which may explain the difference in arch stability between the sexes (43–45)

12* The section for limitation must be revised. Discuss the factors that limit the generalizability of study results.

Thank you for your comment. We clarified as bellow:

The reliance on various static measurement indexes in the included studies significantly restricts the capacity to reach definitive conclusions and apply findings across different clinical contexts. A consensus must be reached on the use of a standardized static measurement index to improve comparability and generalizability. Moreover, neglecting to account for sex and pain duration in the data analysis of some included studies overlooks significant differences in treatment responses, potentially invalidating conclusions for various populations. Researchers should consistently incorporate sex differences and pain duration categories to improve insights and relevance.

Inconsistent measurement methods and terminology (e.g., pes planus, pronation, ND, ankle ROM) create ambiguity that complicates result comparisons across studies, undermining the reliability of evidence. Standardizing biomechanical factors and terminology is crucial for enhancing clarity and facilitating reliable comparisons.

13* Page 12, lines 285-287: “Considering the complex and multifactorial nature of the relationship between lower extremity alignment and LBP, clinicians should take alignments and targeted interventions into account in the management of LBP.” Based on the results of this review there were no causal relationship between alignment and LBP. Please revise.

Thank you for your comment. We clarified as bellow:

Considering the complex and multifactorial nature of the association between lower extremity (such as ankle and hip) alignments and LBP, clinicians should still consider lower extremity alignment as one of important factors in the management of LBP.

14* The implications of the findings of the study for future research should also be clearly stated.

Thank you for your comment. We edited as bellow:

Future research should explore the role of lower extremity alignment in managing low back pain, optimize interventions, and investigate longitudinal studies for sustained improvements. Additionally, it should consider assessing dynamic alignment alongside static alignment for personalized treatment strategies.

---

## [Decision Letter · Decision Letter 1]

30 Aug 2024

PONE-D-24-23407R1Association between lower limb alignment and low back pain: A systematic review with meta-analysisPLOS ONE

Dear Dr. Abbasi,

Thank you for submitting your manuscript to PLOS ONE. After careful consideration, we feel that it has merit but does not fully meet PLOS ONE’s publication criteria as it currently stands. Therefore, we invite you to submit a revised version of the manuscript that addresses the points raised during the review process.

We look forward to receiving your revised manuscript.

Kind regards,

Renato S. Melo, PhD

Academic Editor

PLOS ONE

Journal Requirements:

Reviewers' comments:

Reviewer's Responses to Questions

**Comments to the Author**

1. If the authors have adequately addressed your comments raised in a previous round of review and you feel that this manuscript is now acceptable for publication, you may indicate that here to bypass the “Comments to the Author” section, enter your conflict of interest statement in the “Confidential to Editor” section, and submit your "Accept" recommendation.

Reviewer #1: (No Response)

Reviewer #2: (No Response)

2. Is the manuscript technically sound, and do the data support the conclusions?

Reviewer #1: Yes

Reviewer #2: (No Response)

3. Has the statistical analysis been performed appropriately and rigorously? 

Reviewer #1: Yes

Reviewer #2: (No Response)

4. Have the authors made all data underlying the findings in their manuscript fully available?

Reviewer #1: No

Reviewer #2: (No Response)

5. Is the manuscript presented in an intelligible fashion and written in standard English?

Reviewer #1: Yes

Reviewer #2: (No Response)

6. Review Comments to the Author

Reviewer #1: The authors updated search strategy. But they did not provide the information about the data obtained in the result section. Please add it.

Reviewer #2: (No Response)

7. PLOS authors have the option to publish the peer review history of their article (what does this mean?). If published, this will include your full peer review and any attached files.

Reviewer #1: No

Reviewer #2: No

---

## [Author Response · Author response to Decision Letter 1]

3 Sep 2024

Dear editor and reviewers;

We thank you for your insightful and constructive suggestions. Based on your helpful suggestions, we were able to further improve our manuscript. We carefully considered and addressed all your specific comments and revised the text if necessary. 

Please find your comments in bold and our responses in italic font.

Journal Requirements:

1* Please review your reference list to ensure that it is complete and correct. If you have cited papers that have been retracted, please include the rationale for doing so in the manuscript text, or remove these references and replace them with relevant current references. Any changes to the reference list should be mentioned in the rebuttal letter that accompanies your revised manuscript. If you need to cite a retracted article, indicate the article’s retracted status in the References list and also include a citation and full reference for the retraction notice.

Thank you for bringing this important issue to our attention. We have thoroughly reviewed all our references, considering the significance of the point you raised We confirm that none of refrences have been retracted. Therefore, there is no need to change the references or provide an explanation regarding the use of retracted references in the manuscript.

Reviewer #1

 Methods

1* The authors updated search strategy. But they did not provide the information about the data obtained in the result section. Please add it.

Thank you for your comment. 

We have updated the search strategy through September 2024, and the flowchart has been revised accordingly. based on the screening, no papers were added to our review.

---

## [Decision Letter · Decision Letter 2]

8 Sep 2024

Association between lower limb alignment and low back pain: A systematic review with meta-analysis

PONE-D-24-23407R2

Dear Dr. Mousavi,

We’re pleased to inform you that your manuscript has been judged scientifically suitable for publication and will be formally accepted for publication once it meets all outstanding technical requirements.

Kind regards,

Renato S. Melo, PhD

Academic Editor

PLOS ONE

Additional Editor Comments (optional):

Reviewers' comments:

Reviewer's Responses to Questions

**Comments to the Author**

1. If the authors have adequately addressed your comments raised in a previous round of review and you feel that this manuscript is now acceptable for publication, you may indicate that here to bypass the “Comments to the Author” section, enter your conflict of interest statement in the “Confidential to Editor” section, and submit your "Accept" recommendation.

Reviewer #1: All comments have been addressed

2. Is the manuscript technically sound, and do the data support the conclusions?

Reviewer #1: Yes

3. Has the statistical analysis been performed appropriately and rigorously? 

Reviewer #1: Yes

4. Have the authors made all data underlying the findings in their manuscript fully available?

Reviewer #1: Yes

5. Is the manuscript presented in an intelligible fashion and written in standard English?

Reviewer #1: Yes

6. Review Comments to the Author

Reviewer #1: The authors addressed my comment accurately. This manuscript is appropriate for publication in the present version.

7. PLOS authors have the option to publish the peer review history of their article (what does this mean?). If published, this will include your full peer review and any attached files.

Reviewer #1: No

---

## [Editor Report · Acceptance letter]

1 Oct 2024

PONE-D-24-23407R2 

PLOS ONE

Dear Dr. Mousavi, 

I'm pleased to inform you that your manuscript has been deemed suitable for publication in PLOS ONE. Congratulations! Your manuscript is now being handed over to our production team.

Kind regards, 

on behalf of

Dr. Renato S. Melo 

Academic Editor

PLOS ONE